# Simplified 0+1 and 1+1 pneumococcal vaccine schedules in Ho Chi Minh City, Vietnam: protocol for a randomised controlled trial

Beth Temple [1,2] Hau Phuc Tran,[3] Vo Thi Trang Dai,[4] Kathryn Bright,[2] Doan Y Uyen,[3] Anne Balloch,[2] Paul Licciardi,[2,5] Cattram Duong Nguyen [2,5] Catherine Satzke,[2,5] Heidi Smith-Vaughan,[6] Thuong Vu Nguyen,[3] Kim Mulholland[2,7]

For numbered affiliations see end of article.

**Correspondence to**
Beth Temple;
beth.temple@menzies.edu.au

## ABSTRACT

**Introduction** Reduced-dose schedules offer a more efficient and affordable way to use pneumococcal conjugate vaccines (PCVs). Such schedules rely primarily on the maintenance of herd protection. The Vietnam Pneumococcal Trial II (VPT-II) will evaluate reduced-dose schedules of PCV10 and PCV13 utilising an unvaccinated control group. Schedules will be compared in relation to their effect on nasopharyngeal carriage and immunogenicity.

**Methods and analysis** VPT-II is a single-blind open-label randomised controlled trial of 2500 infants in three districts of Ho Chi Minh City, Vietnam. Eligible infants have no clinically significant maternal or perinatal history and are born at or after 36 weeks' gestation. Participants are recruited at 2 months of age and randomly assigned (4:4:4:4:9) using block randomisation, stratified by district, to one of five groups: four intervention groups that receive PCV10 in a 0+1 (at 12 months) or 1+1 (at 2 and 12 months) schedule or PCV13 in the same 0+1 or 1+1 schedule; and a control group (that receives a single dose of PCV10 at 24 months). Participants are followed up to 24 months of age. The primary outcome is vaccine-type pneumococcal carriage at 24 months of age. Secondary outcomes are carriage at 6, 12 and 18 months of age and the comparative immunogenicity of the different schedules in terms of antibody responses, functional antibody responses and memory B cell responses.

**Ethics and dissemination** Ethical approval has been obtained from the Human Research Ethics Committee of the Royal Children's Hospital Melbourne and the Vietnam Ministry of Health Ethics Committee. The results, interpretation and conclusions will be presented to parents and guardians, at national and international conferences and published in peer-reviewed open access journals.

**Trial registration number** NCT03098628.

## Strengths and limitations of this study

► Conducted in a country without routine pneumococcal conjugate vaccine (PCV) use, allowing inclusion of an unvaccinated control group and measurement of the reduction in carriage afforded by the reduced-dose schedules.

► Includes 0+1 and 1+1 schedules of both PCV10 and PCV13, allowing a head-to-head comparison of these two vaccines.

► Utilises molecular methods to provide a detailed assessment of the effect of reduced-dose schedules on pneumococcal carriage and density at key post-vaccination time points.

► Also includes a full range of immunological outcome measures, encompassing antibody responses (by ELISA), functional antibody responses (by opsono-phagocytic assay) and memory B cell responses.

► Does not include an accepted (WHO-recommended) schedule as an additional comparator group.

(PCVs) currently in use, PCV10 (*Synflorix*, GlaxoSmithKline) and PCV13 (*Prevnar-13*, Pfizer). A third PCV, *Pneumosil* (10-valent, Serum Institute of India), received WHO prequalification in December 2019. Both PCV10 and PCV13 are available through the Advanced Market Commitment mechanism, a vaccine purchase process developed by Gavi to support vaccine introduction into low-income and middle-income countries. Under this mechanism, countries pay a gradually increasing share of the cost of their Gavi-supported vaccines. Countries that have introduced PCVs with Gavi support are rapidly approaching the time when they will have to pay most, if not all, of the price of the vaccine, necessitating simpler, less expensive ways of using PCVs.

Introduction of PCV has been associated with dramatic reductions in pneumococcal disease.[2–4] The benefits of vaccination are

## INTRODUCTION
### Background and rationale

*Streptococcus pneumoniae* (the pneumococcus) causes significant morbidity and mortality in children under 5 years of age.[1] There are two infant pneumococcal conjugate vaccines

not only seen among vaccinated individuals (direct protection), but also in the wider unvaccinated population (indirect herd protection) through reduced nasopharyngeal (NP) carriage and transmission of vaccine type (VT) pneumococci.[5] The manufacturers recommend a 3+1 schedule (a three-dose primary series with a booster), but WHO currently recommends a three-dose schedule (either 3+0, a three-dose primary series without a booster, or 2+1, a two-dose primary series with a booster).[6] There is evidence to suggest that the number of doses could be further reduced with schedules designed to maintain herd protection. The UK, a country with established herd immunity from a mature PCV programme in a 2+1 schedule, recently became the first country to move to a 1+1 reduced-dose schedule. That decision was based on favourable postbooster immunogenicity compared with a 2+1 schedule, with equivalent or superior antibody levels following a 1+1 schedule for nine of the 13 serotypes in PCV13.[7]

In a 1+1 schedule, the first dose is likely to confer some protection to the recipient, and importantly provides priming for the later booster dose. A single dose of PCV7 showed 73% effectiveness against invasive pneumococcal disease (IPD) during a period of vaccine shortage in the USA,[8] and a single dose of PCV13 in the UK showed 60% effectiveness against IPD from the additional serotypes not included in PCV7.[9] A single dose of PCV in infancy also generates a measurable and significant immune response,[10–13] and is better than multiple doses at priming for a booster dose for some serotypes.[7 14] The purpose of the second dose in a 1+1 schedule is maintenance of herd protection. The timing of this dose should consider maximising individual protection of the recipient (through earlier administration) and optimising protection against carriage (through later administration); 12 months of age offers a balance between these two factors. In Vietnam, this also coincides with the first routine Japanese Encephalitis Vaccine visit. A further simplified schedule is a 0+1 schedule, involving only the second dose from a 1+1 schedule with no primary immunisation. The rationale is that a single dose will be sufficient to maintain pre-existing herd protection and control the potential re-emergence of VT, while recognising the reduced individual protection during the first year of life.

Reduced-dose (0+1 or 1+1) schedules could be implemented in countries with established PCV programmes, or they could be introduced as a primary course in their own right (in conjunction with a comprehensive catch-up campaign). Such simplified regimens present an opportunity to use existing PCVs more efficiently, and to make pneumococcal vaccination more affordable. The Vietnam Pneumococcal Trial II (VPT-II) will evaluate both PCV10 and PCV13 in a 0+1 schedule at 12 months of age and a 1+1 schedule at 2 and 12 months of age, in a largely PCV-naïve population utilising an unvaccinated control group, to assess their effect on NP carriage and immunogenicity during the first 2 years of life.

## Explanation for choice of comparators

PCV is not part of the routine infant vaccination programme in Vietnam. Inclusion of a control group that receives no infant doses of PCV is therefore justified. PCV13 was not licensed in Vietnam at the time the trial started, and PCV10 was available on the private market but not widely used. Control group participants receive a single dose of PCV10 at 24 months of age in order that all trial participants receive the benefit of pneumococcal vaccination, regardless of group allocation.

## Objectives

The overall objective of this trial is to fully evaluate the microbiological and immunological effects of 0+1 and 1+1 schedules for both PCV10 and PCV13.

### Coprimary objectives

► To evaluate the effect of a 1+1 PCV schedule on NP carriage during the first 2 years of life, comparing (A) PCV10-vaccinated and (B) PCV13-vaccinated participants with unvaccinated controls

► To evaluate the effect of a 0+1 PCV schedule on NP carriage during the first 2 years of life, comparing (A) PCV10-vaccinated and (B) PCV13-vaccinated participants with unvaccinated controls

### Secondary objectives

► To evaluate the immunogenicity of a 1+1 schedule of PCV10 or PCV13.

► To evaluate the immunogenicity of a 0+1 schedule of PCV10 or PCV13.

► To determine whether the response to a dose of PCV10 or PCV13 at 12 months of age is enhanced by a dose at 2 months of age (ie, 0+1 vs 1+1).

► To investigate the kinetics of the immune response to PCV vaccination at 12 months of age, comparing antibody levels and memory B cell responses 7 days and 28 days postvaccination.

► To examine the serotype profile of transferred maternal pneumococcal antibodies at 2 months of age.

► To directly compare PCV10 and PCV13 in terms of immune responses and effect on NP carriage.

## Trial design

VPT-II is a single-blind, open-label, randomised controlled phase II/III clinical trial to investigate simplified PCV schedules that are focused primarily on the generation of herd immunity. Participants are randomised to one of five groups: group V receives a 0+1 PCV10 schedule, group W receives a 0+1 PCV13 schedule, group X receives a 1+1 PCV10 schedule, group Y receives a 1+1 PCV13 schedule and group Z is a control group (that receives a single dose of PCV10 at 24 months of age). The trial registration data set can be found in online supplemental appendix 1.

## METHODS

### Study setting

The trial is conducted in three districts within Ho Chi Minh City, Vietnam (districts 4, 7 and 8). Districts are divided into communes, each of which has a health centre that provides preventive health services including EPI vaccines, along with primary healthcare services. The trial is conducted in one commune health centre in each district, with participants drawn from the surrounding communes within that district.

### Eligibility criteria

#### Inclusion criteria

In order to be eligible, subjects must meet all of the following criteria: aged between 2 months and 2 months plus 2 weeks, no significant maternal or perinatal history, born at or after 36 weeks gestation, written and signed informed consent from parent/legal guardian, lives within approximately 30 min of the commune health centre, and family anticipates living in the study area for the next 22 months.

#### Exclusion criteria

Subjects meeting any of the following criteria will not be eligible to participate: known allergy to any component of the vaccine, allergic reaction or anaphylactic reaction to any previous vaccine, known immunodeficiency disorder, known HIV-infected mother, known thrombocytopenia or coagulation disorder, administration or planned administration of any immunoglobulin or blood product since birth, severe birth defect requiring ongoing medical care, chronic or progressive disease, seizure disorder, history of severe illness, receipt of any 2 months vaccines through the EPI programme, or family plans on giving the infant *Quinvaxem* (diphtheria, tetanus, pertussis (DTP) *Haemophilus influenzae* type b, hepatitis B vaccine).

### Interventions

The interventions are PCV10 and PCV13. There are four intervention groups, which receive: PCV10 at 12 months of age (0+1 PCV10, group V), PCV13 at 12 months of age (0+1 PCV13, group W), PCV10 at 2 and 12 months of age (1+1 PCV10, group X), or PCV13 at 2 and 12 months of age (1+1 PCV13, group Y). Control group participants (group Z) receive a single dose of PCV10 at 24 months of age. PCV is administered by intramuscular injection into the anterolateral thigh in children less than 18 months old and in the deltoid muscle of the arm in children aged 18 months and over. All vaccinations are performed by Ministry of Health nurses specifically trained in infant vaccine administration. Single-dose vials of PCV10 and single-dose prefilled syringes of PCV13 are used.

### Criteria for discontinuing or modifying allocated interventions

There is no modification of doses for participants in this study. If a participant has an allergic or anaphylactic response to vaccination, they will be withdrawn from the study. Participants may also be withdrawn voluntarily by the parent/legal guardian at any time, or by the study staff if they refuse any further study procedures or develop any of the exclusion criteria during the course of the study.

### Strategies to improve and monitor adherence

Scheduled visit dates are noted on a health record card kept by the parent. If a participant does not attend a scheduled visit, a reminder phone call is made from the study clinic. If the participant cannot be contacted directly, their local Commune Health Centre is contacted for further follow-up by phone or by home visit.

### Relevant concomitant care

Participants also receive four doses of *Infanrix-hexa*, which is a popular choice of DTP-containing vaccine but is only available on the private market, instead of the routine EPI vaccine *Quinvaxem*. With the exception of *Quinvaxem*, other vaccines are permitted in this study providing there are 2 weeks between the administration of other vaccines and study vaccines. Other medications are also permitted, with the exception of immunosuppressive medication and medications listed as contraindicated to the study vaccines.

### Outcomes

#### Primary outcome measure

The primary outcome measure to address each of the coprimary objectives is carriage of VT pneumococci, defined as carriage of serotypes 1, 4, 5, 6B, 7F, 9V, 14, 18C, 19F or 23F for the PCV10 groups, or carriage of serotypes 1, 3, 4, 5, 6A, 6B, 7F, 9V, 14, 18C, 19A, 19F or 23F for the PCV13 groups. The VT pneumococcal carriage prevalence (defined as the proportion of participants positive for VT pneumococcal carriage) will be determined at 24 months of age (primary endpoint), and also at 6, 12 and 18 months of age (secondary endpoints). Traditional culture methods (colonial morphology, α-haemolysis and the optochin test) and serotyping by latex agglutination/Quellung, with *lytA* PCR confirmation of nonencapsulated isolates, will be the main methods used to analyse NP swabs collected at 6 and 12 months of age. Quantitative real-time PCR (qPCR) targeting the *lytA* gene[15] and serotyping by microarray[16] will be the main methods used to analyse NP swabs collected at 18 and 24 months of age.

#### Secondary microbiological outcome measures

► Carriage of any pneumococcal serotype at 6, 12, 18 and 24 months of age.
► Non-VT pneumococcal carriage at 6, 12, 18 and 24 months of age.
► Serotype-specific pneumococcal carriage at 6, 12, 18 and 24 months of age.
► Density of pneumococcal carriage (overall, VT, non-VT and serotype-specific) at 18 and 24 months of age.

#### Secondary immunological outcome measures

► Serotype-specific IgG antibody concentrations for all PCV13 serotypes, measured by ELISA[17] from all blood samples.

**Table 1** Schedule of enrolment, interventions and assessments

| Age (months) | 2 months | 3 months | 4 months | 6 months | 12 months | 18 months | 24 months |
|---|---|---|---|---|---|---|---|
| **Enrolment** | | | | | | | |
| Informed consent | ✓ | | | | | | |
| Eligibility assessment | ✓ | | | | | | |
| Allocation | ✓ | | | | | | |
| **Interventions** | | | | | | | |
| PCV10 | X | | | | V, X | | Z |
| PCV13 | Y | | | | W, Y | | |
| Infanrix-hexa | ✓ | ✓ | ✓ | | | ✓ | |
| **Assessments** | | | | | | | |
| Demographics | ✓ | | | | | | |
| Household characteristics | ✓ | | | | | | |
| Nasopharyngeal swab | | | | V-Y | V-Y | ✓ | ✓ |
| General health | ✓ | ✓ | ✓ | ✓ | ✓ | ✓ | ✓ |

✓Indicates applies to all groups (V–Z), otherwise group(s) specified.
PCV, pneumococcal conjugate vaccine.

► Opsonisation indices (OI) for all PCV13 serotypes, measured by opsonophagocytic assay (OPA)[18] for 50 participants per intervention group (groups V–Y) pre and 4 weeks post-12 months dose of PCV.

► Polysaccharide-specific memory B cells for serotypes 1, 5, 6A, 6B, 14, 19A and 23F, enumerated by ELISPOT[19] for 50 participants per intervention group (groups V–Y) pre, 7 days post and 28 days post-12 months dose of PCV and at 24 months of age.

An overview of the procedures for collection, transportation and laboratory analyses of the blood and NP samples can be found in online supplemental appendix 2.

### Participant timeline

Participants are enrolled at 2 months of age and followed up to 24 months of age (table 1). Group V–Y participants provide NP swabs at 6 and 12 months of age, and all participants provide NP swabs at 18 and 24 months of age for analysis of the NP carriage outcomes. A subset of 200 participants per group will be included in the immunology substudy and provide three (groups V–Y) or one (group Z) blood sample(s) over the course of the trial for analysis of vaccine responses (table 2).

### Sample size

The target sample size is 2500 with an allocation ratio of 4:4:4:4:9. This allocation ratio provides the greatest power for a total sample size of 2500 and results in target group sizes of 400 for each of the four different infant vaccination schedules (groups V–Y) and 900 for the control group (group Z). Sample size calculations are based on the primary outcome of effect on VT pneumococcal carriage at 24 months of age. For each of the four vaccination schedules compared with the unvaccinated control group, the proposed sample size provides 82% power to detect a 40% reduction in the vaccinated group, with a two-sided type I error rate of 5%. The sample size calculations assume 15% VT pneumococcal carriage among controls, based on data from our previous pneumococcal vaccine trial in Ho Chi Minh City,[20] and 10% lost to follow-up.

### Recruitment

Participants are recruited from infants born in the study communes during the enrolment period. Potential participants are identified from commune health centre birth records and are visited by commune health centre staff when they are approximately 6 weeks old. Verbal and written information about the trial is provided in Vietnamese and those interested in participating are referred to the study clinic when the infant is approximately 2 months old. At this time, written informed consent is obtained (online supplemental appendix 3), after which a study nurse/doctor examines the infant to ensure that all the eligibility criteria are met. Recruitment rates will be monitored on a monthly basis and meetings held with study staff and commune health centre staff to discuss any significant declines in recruitment rates.

### Allocation

The allocation ratio of Groups V, W, X, Y and Z is 4:4:4:4:9. The first 200 participants enrolled into each of groups V–Y, and a randomly selected 200 of the first 450 participants concurrently enrolled into group Z, will go into the immunology sub-study. These participants (with the exception of group Z) will be further randomised into one of four subgroups (a, b c or d) in an equal allocation ratio (table 2). Randomisation will be conducted by a database manager in Australia, using a computer-generated

**Table 2** Schedule of blood samples for immunology substudy (subset of groups V–Z)

| Age (months) | 2 months | 3 months | 12 months | Post-12 months* | 24 months |
|---|---|---|---|---|---|
| Subgroup a | | | V-Y† | V-Y | V-Y |
| Sample volume | | | 7.5 mL | 2 mL | 3.5 mL |
| Assays | | | ELISA, B cell | ELISA | ELISA |
| Subgroup b | V | W-Y | V-Y | V-Y | |
| Sample volume | 2 mL | 2 mL | 2 mL | 7.5 mL | |
| Assays | ELISA | ELISA | ELISA | ELISA, B cell | |
| Subgroup c | | | V-Y | V-Y | V-Y |
| Sample volume | | | 3.5 mL | 7.5 mL | 3.5 mL |
| Assays | | | ELISA | ELISA, B cell | ELISA |
| Subgroup d | | | V–Y | V–Y | V–Y |
| Sample volume | | | 3.5 mL | 3.5 mL | 7.5 mL |
| Assays | | | ELISA, OPA‡ | ELISA, OPA | ELISA, B cell |
| Subgroup Z-I | | | | | Z |
| Sample volume | | | | | 3.5 mL |
| Assays | | | | | ELISA |

*The post-12 months blood sample is collected at 12 months plus 7 days in subgroups a and b and at 12 months plus 28 days in subgroups c and d.
†3.5 mL sample for ELISA only in group W.
‡OPAs not performed in group W.
OPA, opsonophagocytic assay.

list of random numbers in a block randomisation scheme stratified by district. The group allocation, and subgroup allocation where relevant, is contained within a sealed envelope at the study clinic. Participants are assigned to a study group by a study doctor, using the next available envelope. The envelopes are prepared and sealed in the study office in Vietnam, by staff with no involvement in the recruitment process.

### Blinding
All outcome measures are laboratory based. Laboratory staff are blinded to the study group allocation. Laboratory samples are labelled with a unique ID number, which does not identify the study group. Given the different timing of the vaccination schedules in the different groups, the study nurses, vaccine administrators and participants will not be blinded to the study group allocation.

### Data collection methods
Data collected at the clinic are documented by dedicated, trained study staff using standardised forms. Blood samples and NP swabs are collected by staff specifically trained in the collection of samples from infants, and the volume of blood and swab quality are recorded. Laboratory data generated in both Vietnam and Australia are entered directly into dedicated databases in the laboratories by the laboratory staff and sent periodically to the data management team in Australia.

### Retention
Appointments are documented on a parent-held health record card and a reminder phone call made the week before the scheduled visit. Missed visits are rebooked by phone and participants who miss a study visit will continue to be followed up for both sample collection and vaccine administration where possible. Participants receive a small payment at each visit towards transport costs.

### Data management
Data collected at the clinic are entered on-site into a secure, web-based Research Electronic Data Capture (REDCap) system.[21] All entered data are monitored against the source documents for accuracy and completeness, and a series of data checks are performed on a regular basis to identify potential errors in the data. An audit trail of corrections and changes to the data is stored within the REDCap database. Immunology results are double-entered in a Microsoft Excel spreadsheet. NP culture results are entered in a Microsoft Access database and qPCR and microarray results are exported from SentiNET into Microsoft Excel or REDCap databases. The data collection forms and laboratory results are linked at the time of analysis.

### Statistical methods
#### Analysis of primary and secondary outcomes
The primary outcome of VT carriage will be presented as a proportion (prevalence). VT carriage prevalence in each of the vaccinated groups (PCV10-type carriage for

groups V and X and PCV13-type carriage for groups W and Y) will be compared with unvaccinated controls. Prevalence ratios (ratio of prevalence in each vaccinated group to prevalence in unvaccinated controls) with 95% CIs will be calculated, and groups compared using $\chi^2$ tests (two-sided p values reported). The primary endpoint is 24 months of age, with secondary endpoints of 6, 12 and 18 months of age. At 6 and 12 months of age, time points at which no swabs were collected from group Z, swabs from the 0+1 groups (groups V and W combined) will form the unvaccinated comparator.

Immunological outcomes will be summarised in terms of: geometric mean concentrations (GMCs) and the proportion with protective antibody levels ($\geq 0.35\,\mu g/mL$)[22] with 95% CIs for serotype-specific IgG data, geometric mean OIs (GMOIs) and the proportion with an OI $\geq 8$[23] with 95% CIs for OPA data, and the mean number of antibody secreting cells per $10^6$ PBMCs for polysaccharide-specific memory B cell data. GMC and GMOI ratios and risk differences with 95% CIs will also be calculated. Means will be compared using t-tests and proportions compared using Fisher's exact tests.

### Populations of analysis
Analysis is planned on an intention-to-treat population, with all participants to be analysed in the group to which they were randomised. Withdrawn participants may not contribute data at all time points as blood and NP samples may not be collected after their withdrawal.

### Data monitoring
#### Data monitoring committee
Safety oversight is under the direction of an independent data safety and monitoring board (DSMB), in accordance with a DSMB Charter kept in the trial office. The DSMB will consist of at least three members, including two physicians and one biostatistician. The DSMB will meet periodically to review aggregate and individual participant data related to safety, data integrity and overall conduct of the trial, including a detailed review of all serious adverse events (SAEs).

#### Interim analyses and stopping guidelines
No interim analyses are planned. Statistical rules will not be used to halt study enrolment or vaccine administration. Stopping guidelines are based on safety. An extraordinary meeting of the DSMB will be called in the event that serious safety issues emerge, to provide recommendations regarding termination of the trial. A final decision to terminate rests with the principal investigators and the sponsor.

### Harms
Data on SAEs will be collected throughout the duration of the main study, with parents asked about hospitalisations and significant signs and symptoms at each study visit and through a regular review of hospital records. All SAEs will be recorded on the standard Vietnam Ministry of Health reporting form and reported to the principal

investigators and the ethics committees. Participants will be kept under observation for 30 min following vaccine administration to monitor for any adverse reactions, and adverse events that may contraindicate further vaccinations will be assessed following all vaccination visits. Reactogenicity will be assessed following all doses of PCV using parent held diary cards.

### Auditing
External site monitoring will be provided by Family Health International (FHI360), to independently assess protocol and good clinical practice (GCP) compliance. The frequency of monitoring visits and the level of detail of the monitoring will be documented in a contract between FHI360 and the sponsor. Monitoring visits will start prior to enrolment of the first participant and continue through to study-close-out.

### Patient and public involvement
Patients were not involved in the development, design, recruitment or conduct of the study. Community consultation took place at the district level during the design phase, as well as discussion and approval of the design from the district health centres, the city Department of Health, and the People's Committee of Ho Chi Minh City. Participants will be informed of the overall study results by email or by post, with addresses collected at the final study visit.

## ETHICS AND DISSEMINATION
### Research ethics approval
The protocol and the informed consent form (ICF) have approval from the Institutional Review Board at the Pasteur Institute of Ho Chi Minh City, the Vietnam Ministry of Health Ethical Review Committee for Biomedical Research and the Human Research Ethics Committee of the Royal Children's Hospital, Melbourne. Both ethics committees receive annual reports on the trial progress, for continuing approval of the trial.

### Protocol amendments
Any modifications to the protocol that may impact on the conduct of the study will be documented in a formal protocol amendment and approved by both ethics committees prior to implementation of the changes. The modified protocol will be given a new version number and date. The ethics committees will also be notified of any minor corrections/clarifications or administrative changes to the protocol, which will be documented in a protocol amendment letter. Significant protocol changes will also be updated in the ClinicalTrials.gov record.

### Consent
#### Obtaining consent
The consent process is undertaken by specifically trained study staff. The study staff go through the ICF, translated into Vietnamese, in detail with the potential participant's parent/legal guardian. Study staff then discuss the trial further and answer any questions that may arise. Written

informed consent is required prior to enrolment of the infant into the study and is provided by the parent/legal guardian as the participants are too young to provide consent themselves. A copy of the ICF will be given to the parent/legal guardian for their records.

## Ancillary studies

Specific consent for the indefinite storage of blood and NP samples for future research related to the trial will be obtained from the parent/legal guardian and recorded on the ICF. Any future research will undergo ethical review. Any samples for which indefinite storage is not consented to will be destroyed at the close of the study.

## Confidentiality

All study-related information will be stored securely and held in strict confidence. All documents kept at the study clinics are stored in locked cabinets. The REDCap database is password protected. All documents maintained centrally are stored in the trial office, which is kept locked. The laboratory samples and electronic laboratory data are coded by participant ID number and do not contain names. Access to participants' information will be granted to FHI360 for monitoring purposes, and to the Ethics Committees or DSMB if required.

## Declaration of interests

KM, CS and CDN are investigators on a collaborative study on PCV impact on adult pneumonia funded by Pfizer.

## Access to data

The final dataset will be under the custody of the trial sponsor, MCRI. The Principal Investigator, trial manager and trial statistician will have access to the full anonymised final dataset.

## Ancillary and post-trial care

Participants are advised to come to the study clinic for ancillary care, or to Children's Hospital Number 1 or Children's Hospital Number 2 in Ho Chi Minh City, where they will not be charged for treatment and services. All participants are covered by clinical trials insurance for trial related harms.

## Dissemination policy
### Plans

Following completion of the trial, the results will be submitted for publication in peer-reviewed journals and presented at relevant international conferences. The results will be disseminated regardless of the magnitude or direction of effect. This research is undertaken as a collaboration between MCRI and the Pasteur Institute of Ho Chi Minh City. Either party must obtain the prior approval of the other party in advance of submitting a manuscript for publication, and such approval will not be unreasonably withheld.

### Authorship

A small group of senior investigators will consider all proposed publications, with the final decision on content and authorship resting with the principal investigator. The role of each author will be published in line with journal requirements.

Group authors may be used where appropriate. There are no plans for the use of professional writers.

## Reproducible research

Data will be made publicly available in accordance with the rules set out by the Bill & Melinda Gates Foundation.

**Author affiliations**

[1]Global and Tropical Health Division, Menzies School of Health Research, Charles Darwin University, Casuarina, Northern Territory, Australia
[2]Infection and Immunity, Murdoch Children's Research Institute, Parkville, Victoria, Australia
[3]Department for Disease Control and Prevention, Pasteur Institute of Ho Chi Minh City, Ho Chi Minh City, Viet Nam
[4]Department of Microbiology and Immunology, Pasteur Institute of Ho Chi Minh City, Ho Chi Minh City, Viet Nam
[5]Department of Paediatrics, University of Melbourne, Melbourne, Victoria, Australia
[6]Child Health Division, Menzies School of Health Research, Charles Darwin University, Darwin, Northern Territory, Australia
[7]Epidemiology and Population Health, London School of Hygiene & Tropical Medicine, London, UK

**Correction notice** This article has been corrected since it was published. The surname of the last author has been corrected to Mulholland.

**Acknowledgements** We thank the study participants and their families and the study staff. We acknowledge the contributions of Phan Trong Lan, Sophie La Vincente, Helen Thomson and Thi Que Huong Vu (deceased).

**Contributors** BT wrote the first draft of this manuscript and drafted the trial protocol with input from HPT, KB and DYU. HPT also advised on the study design and oversaw the approval processes in Vietnam. VTTD, AB, PL, CS and HS-V were involved in the laboratory-related aspects of the design and protocol development. CDN advised on the study design and statistical aspects of the trial. TVN was involved in the design and establishment of the trial and had overall responsibility for its conduct in Vietnam as the site Principal Investigator. KM conceived the study, provided oversight for all aspects of the design and implementation, and had overall responsibility for the trial as Principal Investigator. All authors contributed to refinement of the trial protocol and reviewed and approved this manuscript.

**Funding** This work is supported by the Bill & Melinda Gates Foundation (grant number OPP-1116833/INV-008627). We also acknowledge the Victorian Government's Operational Infrastructure Support Programme.

**Competing interests** All authors receive salary support from grants from the Bill & Melinda Gates Foundation. KM, CS and CDN are investigators on a collaborative study on PCV impact on adult pneumonia funded by Pfizer.

**Patient consent for publication** Not applicable.

**Provenance and peer review** Not commissioned; externally peer reviewed.

**ORCID iDs**
Beth Temple http://orcid.org/0000-0002-4885-9848
Cattram Duong Nguyen http://orcid.org/0000-0002-0599-8645

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
