## [Reviewer comments · BMJ Open]

ARTICLE DETAILS

TITLE (PROVISIONAL)	Protocol for a randomised controlled trial of simplified 0+1 and 1+1 pneumococcal vaccine schedules in Ho Chi Minh City, Vietnam
AUTHORS	Temple, Beth; Tran, Hau; Dai, Vo Thi Trang; Bright, Kathryn; Uyen, Doan; Balloch, Anne; Licciardi, Paul; Nguyen, Cattram; Satzke, Catherine; Smith-Vaughan, Heidi; Vu Nguyen, Thuong; Muholland, Kim

VERSION 1 – REVIEW

REVIEWER	Zhang, Tao Fudan University
REVIEW RETURNED	02-Sep-2021

GENERAL COMMENTS	This study intends to test whether the reduced-dose schedules of PCVs provides similar protection on the pneumococcal carriage at 24 months of age. This is a well-designed single-blind randomized controlled trial, and the proposal is well written. I only have some minor concerns about the proposal. The authors explained the reasons to have the second dose of the 1+1 schedule at 12 months. While no rationale for the 0+1 schedule to receive at 12 months. It's better to provide some evidence. Table 1a, as the primary outcome is carriage, why not collect the nasopharyngeal swab at the time of enrollment—which at 2 months?
---

REVIEWER	Swarthout, Todd UCL, Division of Infection and Immunity
REVIEW RETURNED	14-Sep-2021

GENERAL COMMENTS	The study protocol by Temple et al presents the study design for the Vietnam Pneumococcal Trial II (VPT-II), a single-blind randomised controlled trial that will evaluate reduced-dose schedules of PCV10 and PCV13. Schedules will be evaluated in relation to their effect on NP carriage and immunogenicity, with a primary outcome of vaccine serotype pneumococcal carriage at 24 months of age. The topic of reduced-dose schedules is of importance as a means to evaluate efficient and affordable ways to utilise PCVs, including in settings where PCV is currently not available. It is also relevant to GAVI-funded countries given the importance of indirect protection among PCV-unvaccinated populations in cost-effectiveness and ongoing work to assess appropriate timing to consider transition to a 1+1 vaccine schedule.
---

The paper is well written but there are a number of points that need clarifying or would benefit from a more nuanced discussion. I present these below as major, semi-major and minor comments.

Note: I report using page numbers of the pdf provided. I have used the page number (# of 33) as reported on top left or right of page.

Major

1. **Reviewer:** As this is an RCT, I would like to see a Consort Statement as part of the appendix.

2. Page 8 of 33, Line ~38-45: The author writes “*There is evidence to suggest that the number of doses could be further reduced with schedules designed to maintain herd protection. The UK recently became the first country to move to a 1+1 reduced dose schedule, based on favourable post-booster immunogenicity compared with a 2+1 schedule.*”

Reviewer: The author often refers to the potential importance of a reduced schedule to maintain herd protection. This is reinforced by the authors reference to the UK’s shift to a 1+1. However, Vietnam is PCV-naïve and does not currently benefit from the herd immunity that has been established, in the case of the UK, through a mature PCV programme. While I am very much in favour of evaluating these reduced (<3) dose schedules, there is adequate disconnect between these two settings (UK and Vietnam) that the author should address the question of equipoise in justifying a schedule (0+1 or 1+1) that does not follow either WHO (3+0 or 2+1; page 8 of 33, lines 35-38) or the manufacturer’s (3+1; page 8 of 33, lines 33-35) recommendation. This could include an epidemiological, an ecological or a strictly financial justification.

3. Page 9 of 33, Line 8-19: The author writes: “*The implication is that a single dose will be sufficient to maintain pre-existing herd protection and control the potential re-emergence of vaccine types, while recognising the reduced individual protection during the first year of life. Reduced-dose (0+1 or 1+1) schedules could be implemented in countries with established PCV programs, or they could be introduced as a primary course in their own right (in conjunction with a comprehensive catch-up campaign).*”

Reviewer: Similar to my comment above (Major-1), the author needs to justify evaluating a 1- and 2-dose PCV schedule in a PCV-naïve setting (*recognising the reduced individual protection during the first year of life*) and without a catch-up campaign (*could be introduced as a primary course in their own right (in conjunction with a comprehensive catch-up campaign)*). The author appears to be establishing scenarios (herd protection, catch-up campaigns) in which reduced dose PCV schedules could be impactful...without implementing those scenarios or, at minimum, acknowledging the absence of those scenarios.

4. Page 9 of 33, Line 33-35: Author writes: “*Inclusion of a control group that receives no infant doses of PCV is therefore justified.*”

Reviewer: Please clarify what justifies a control group (PCV not being part of the routine infant vaccination program in

Vietnam or that PCV10 is not widely used on the private market...or both?

5. Page 9 of 33, Lines 35-37: Author writes: "*Control group participants receive a single dose of PCV10 at 24 months of age.*"

Reviewer: Please clarify why the study provides PCV10 and not PCV13 to the control group.

Reviewer: Please clarify why the controls receive PCV at the end of the study.

Semi-major

1. As per BMJ Open policy, "*if data collection is complete, we will not consider the manuscript.*" The ClinicalTrials.gov NCT03098628 reports the study as "Active, not recruiting," with 2501 enrolled 2501. On page 6 of 33 the authors report the recruitment period as "8 March 2017 – 11 June 2020."

Reviewer: I would encourage BMJ Open to publish this protocol, despite the BMJ Open policy of not considering the manuscript if data collection is complete. While not wanting to be pedantic, I do feel this late submission detracts from the admirable goal of (as stated by BMJ Open policies) helping prevent unnecessary duplication of work, hopefully enabling collaboration, and increasing transparency.

2. Page 8 of 33, Line ~20-24: The author writes "*The Countries that have introduced PCVs with Gavi support are rapidly approaching the time when they will have to pay most, if not all, of the price of the vaccine,...*"

Reviewer: The author should provide a few more words to clarify the GAVI policy of support...i.e. that GAVI support is not indefinite, after which countries need to transition to alternative funding options, including paying "most, if not all, of the price of the vaccine."

3. Page 8 of 33, Line ~30-23: Author refers to "...but also in the wider population..."

Reviewer: Please clarify if you are referring to the wider unvaccinated population or both vaccinated and unvaccinated.

4. Page 8 of 33, Line ~43-45: The author writes "*...based on favourable post-booster immunogenicity compared with a 2+1 schedule.*"

Reviewer: Please clarify what is meant by “favourable...compared to 2+1. Was it superior, non-inferior, etc?”

Reviewer: I don't mean to be pedantic, but the writing would be more clear/informative for the reader if the author provided more detail. For example, "...compared with the previously implemented 2+1 schedule and after achieving a dramatically reduced VT carriage prevalence.”

5. Page 8 of 33, Line ~49-54: The author writes: “*A single dose of PCV7 showed 73% effectiveness during a period of vaccine shortage...*”

Reviewer: effectiveness against what? Carriage? Disease?

6. Page 9 of 33, Lines 3-6: The author writes “*...12 months of age offers a balance between these two factors.*”

Reviewer: For the reader, please clarify or refer to the evidence that 12m is optimal in this setting. For example, does the disease epidemiology (age of greatest burden of pneumococcal disease) or trends in vaccine uptake justify this age? Also, please report the EPI schedule in Vietnam...is there a routine vaccine visit at 12m of age to optimise uptake (maybe a 3rd factor to consider?)?

7. Page 12 of 33, Lines 14-16: The author writes: “*Participants also receive four doses of Infanrix-hexa...instead of the routine EPI vaccine Quinvaxem.*”

Reviewer: Please explain why this switch (Infanrix-hexa in place of Quinvaxem) is necessary.

8. Page 12 of 33, Lines 16-19: “*Other vaccinations are permitted in this study with a two-week interval from study vaccines.*”

Reviewer: This sentence is not clear. I read it as ‘Other EPI vaccinations are permitted but only if not given within two-weeks of receiving a study vaccine’

9. Page 12 of 33, Lines 39-48: Author writes: *Traditional culture methods...will be the main methods used to analyse NP swabs collected at 6 and 12 months...qPCR targeting the lytA gene and serotyping by microarray will be the main methods used to analyse NP swabs collected at 18 and 24 months of age.*”

Reviewer: Please clarify why you are using more sensitive (microarray) and less specific (lytA) methodologies at the later visits.

Reviewer: Clarify how you will address the fact that microarray increases likelihood of VT detection in co-carriage samples with NVT at high relative abundance and VT at low relative abundance (these samples would likely be defined as NVT using latex but NVT+VT in microarray).

10. Page 12, Line 41: With reference to the use of lytA PCR, I would suggest including a comment on the limitation in using lyA PCR for pneumococcal detection (i.e. low specificity).

Minor

1. Page 4 of 33, Line 28-30; & page 9, line 22: refers to "...schedules as a strategy to maintain herd protection and to make PCVs more affordable

Reviewer: I think this refers to a less costly EPI programme (using 1-2 doses in place of 3-4) and not a less costly PCV vaccine. Please clarify.

2. Page 5 of 33, Line 49/50: refers to "...PCV13 administered at 12 and 2 months..."

Reviewer: Should likely read "...PCV13 administered at 2 and 12 months..."

3. Page 5 of 33, Line 55: reads "*no significant maternal or perinatal history*"

Reviewer: The author should clarify to what type of history is being referred. I suspect it's clinical history but please specify and/or add additional targeted conditions if relevant.

4. Page 11 of 33, Line 44-46: The author writes: "*All vaccinations are performed by nurses specifically trained in infant vaccine administration.*"

Reviewer: Please clarify if these are study-specific or MoH nurses.

5. Page 12 of 33, Lines 56-57: Author writes: "*Serotype-specific pneumococcal carriage at 6, 12, 18 and 24 months of age*"

Reviewer: Is lab team able to determine all VT+NVT serotypes using latex (6 & 12m swabs)? Or should this be "VT serotype specific..."

6. Page 18 of 33, Lines 16-26: Relevant to the section 'Patient and public involvement', is there any intention to disseminate results to the community, via townhall meetings, etc.?

	7. Page 20, Line 26: Relevant to the section 'Authorship', if feasible please clarify how members of the Publication Subcommittee will be selected.
--	---

VERSION 1 – AUTHOR RESPONSE

Reviewer 1

The authors explained the reasons to have the second dose of the 1+1 schedule at 12 months. While no rationale for the 0+1 schedule to receive at 12 months. It's better to provide some evidence.

Response: The provision of the 0+1 schedule at 12 months is to enable direct comparison with the second dose of the 1+1 schedule. We have modified the text to read "A further simplified schedule is a 0+1 schedule, involving only the second dose from a 1+1 schedule with no primary immunization. The rationale is that..." (paragraph 1, page 4)

Table 1a, as the primary outcome is carriage, why not collect the nasopharyngeal swab at the time of enrollment—which at 2 months?

Response: The key outcomes in this RCT involve comparisons between groups. We were limited in the total number of swabs that would be considered acceptable to collect from each participant by the ethics committee in Vietnam. All groups have the same vaccine status at 2 months, and we collected carriage data at 2 months in our previous trial in Ho Chi Minh City, so this time point would not provide new and essential scientific data to justify its inclusion. Of note, in our previous trial we found low carriage at 2 months, with only 3.8% carriage of any pneumococcal serotype.

Reviewer 2

Major point 1: As this is an RCT, I would like to see a Consort Statement as part of the appendix.

Response: We have added the SPIRIT checklist (the checklist for RCT protocols) as a supplementary file

Major point 2: Page 8 of 33, Line ~38-45: The author writes "There is evidence to suggest that the number of doses could be further reduced with schedules designed to maintain herd protection. The UK recently became the first country to move to a 1+1 reduced dose schedule, based on favourable post-booster immunogenicity compared with a 2+1 schedule." Reviewer: The author often refers to the potential importance of a reduced schedule to maintain herd protection. This is reinforced by the authors reference to the UK's shift to a 1+1. However, Vietnam is PCV-naïve and does not currently benefit from the herd immunity that has been established, in the case of the UK, through a mature PCV programme. While I am very much in favour of evaluating these reduced (<3) dose schedules, there is adequate disconnect between these two settings (UK and Vietnam) that the author should address the question of equipoise in justifying a schedule (0+1 or 1+1) that does not follow either WHO (3+0 or 2+1; page 8 of 33, lines 35-38) or the manufacturer's (3+1; page 8 of 33, lines 33-35) recommendation. This could include an epidemiological, an ecological or a strictly financial justification.

Response: We agree that Vietnam is a very different setting than the UK and have highlighted this with the addition of the text "The UK, a country with established herd immunity from a mature PCV

programme in a 2+1 schedule, recently became the first country to move to a 1+1 reduced-dose schedule” (page 3) and “The Vietnam Pneumococcal Trial II will evaluate... in a largely PCV-naïve population...” (page 4). The justification for using these schedules in this trial is that, as a PCV-naïve setting, Vietnam provides the opportunity to evaluate these schedules where the control group is unvaccinated. This enables estimation of the reduction in carriage afforded by these schedules, which is not possible in settings with established PCV programmes. As PCV is not part of the EPI schedule in Vietnam, the schedules in this study will provide benefits over routine care despite not being WHO- or manufacturer-recommended schedules.

Major point 3: Page 9 of 33, Line 8-19: The author writes: “The implication is that a single dose will be sufficient to maintain pre-existing herd protection and control the potential re-emergence of vaccine types, while recognising the reduced individual protection during the first year of life. Reduced-dose (0+1 or 1+1) schedules could be implemented in countries with established PCV programs, or they could be introduced as a primary course in their own right (in conjunction with a comprehensive catch-up campaign).” Reviewer: Similar to my comment above (Major-1), the author needs to justify evaluating a 1- and 2-dose PCV schedule in a PCV-naïve setting (recognising the reduced individual protection during the first year of life) and without a catch-up campaign (could be introduced as a primary course in their own right (in conjunction with a comprehensive catch-up campaign)). The author appears to be establishing scenarios (herd protection, catch-up campaigns) in which reduced dose PCV schedules could be impactful...without implementing those scenarios or, at minimum, acknowledging the absence of those scenarios.

Response: This study will evaluate these schedules in 2500 infants in a trial setting, rather than implement them more widely. This setting provides the opportunity to “utilise an unvaccinated control group” (we have added these words to the last sentence in the Background and rationale, page 4) to generate data comparing children given reduced-dose schedules with unvaccinated controls. Such data will be useful looking to the future at a time either when herd protection has been established with higher-dose schedules or when reduced-dose schedules could be implemented along with a comprehensive catch-up campaign.

Major point 4: Page 9 of 33, Line 33-35: Author writes: “Inclusion of a control group that receives no infant doses of PCV is therefore justified.” Reviewer: Please clarify what justifies a control group (PCV not being part of the routine infant vaccination program in Vietnam or that PCV10 is not widely used on the private market...or both?

Response: We have rearranged the sentences to clarify this (page 4); the text now reads: “PCV is not part of the routine infant vaccination program in Vietnam. Inclusion of a control group that receives no infant doses of PCV is therefore justified.”

Major point 5: Page 9 of 33, Lines 35-37: Author writes: “Control group participants receive a single dose of PCV10 at 24 months of age.” Reviewer: Please clarify why the study provides PCV10 and not PCV13 to the control group. Please clarify why the controls receive PCV at the end of the study.

Response: Control group participants were given PCV10 as “PCV13 was not licensed in Vietnam at the time the trial started” (text added, page 4). Controls receive PCV at the end of the study “in order that all participants receive the benefit of pneumococcal vaccination, regardless of group allocation” (text added, page 4).

Semi-major point 1: As per BMJ Open policy, “if data collection is complete, we will not consider the manuscript.” The ClinicalTrials.gov NCT03098628 reports the study as “Active, not recruiting,” with 2501 enrolled 2501. On page 6 of 33 the authors report the recruitment period as “8 March 2017 – 11 June 2020.” Reviewer: I would encourage BMJ Open to publish this protocol, despite the BMJ Open

policy of not considering the manuscript if data collection is complete. While not wanting to be pedantic, I do feel this late submission detracts from the admirable goal of (as stated by BMJ Open policies) helping prevent unnecessary duplication of work, hopefully enabling collaboration, and increasing transparency.

Response: We thank the reviewer for this comment. Please note that whilst the fieldwork component of the trial is complete, the acquisition of laboratory data is still ongoing.

Semi-major point 2: Page 8 of 33, Line ~20-24: The author writes “The countries that have introduced PCVs with Gavi support are rapidly approaching the time when they will have to pay most, if not all, of the price of the vaccine,...” Reviewer: The author should provide a few more words to clarify the GAVI policy of support...i.e. that GAVI support is not indefinite, after which countries need to transition to alternative funding options, including paying “most, if not all, of the price of the vaccine.”

Response: We have added a sentence “Under this mechanism, countries pay a gradually increasing share of the cost of their Gavi-supported vaccines” (page 3).

Semi-major point 3: Page 8 of 33, Line ~30-23: Author refers to “...but also in the wider population...” Reviewer: Please clarify if you are referring to the wider unvaccinated population or both vaccinated and unvaccinated.

Response: We have added the word “unvaccinated” to clarify this (page 3)

Semi-major point 4: Page 8 of 33, Line ~43-45: The author writes “...based on favourable post-booster immunogenicity compared with a 2+1 schedule.” Reviewer: Please clarify what is meant by “favourable...compared to 2+1. Was it superior, non-inferior, etc? I don’t mean to be pedantic, but the writing would be more clear/informative for the reader if the author provided more detail. For example, “...compared with the previously implemented 2+1 schedule and after achieving a dramatically reduced VT carriage prevalence.”

Response: We have provided additional detail to clarify what is meant by “favourable”, adding the text “with equivalent or superior antibody levels following a 1+1 schedule for nine of the 13 serotypes in PCV13” (page 3)

Semi-major point 5: Page 8 of 33, Line ~49-54: The author writes: “A single dose of PCV7 showed 73% effectiveness during a period of vaccine shortage...” Reviewer: effectiveness against what? Carriage? Disease?

Response: We have added “against IPD” to clarify this (page 3)

Semi-major point 6: Page 9 of 33, Lines 3-6: The author writes “...12 months of age offers a balance between these two factors.” Reviewer: For the reader, please clarify or refer to the evidence that 12m is optimal in this setting. For example, does the disease epidemiology (age of greatest burden of pneumococcal disease) or trends in vaccine uptake justify this age? Also, please report the EPI schedule in Vietnam...is there a routine vaccine visit at 12m of age to optimise uptake (maybe a 3rd factor to consider?)?

Response: We are not stating that 12m is optimal in this setting per se, rather that it is a schedule worth evaluating as it offers a balance between the opposing factors of maximising individual protection (earlier administration) and optimising protection against carriage (later administration), as outlined in the preceding sentence. We have added “In Vietnam, this also coincides with the first routine JEV visit.”

Semi-major point 7: Page 12 of 33, Lines 14-16: The author writes: “Participants also receive four doses of Infanrix-hexa...instead of the routine EPI vaccine Quinvaxem. Reviewer: Please explain why this switch (Infanrix-hexa in place of Quinvaxem) is necessary.

Response: This switch is necessary to ensure that participants don't decline participating in the trial because they want to receive Infanrix-hexa, which is a popular choice of DTP-containing vaccine. We have added this detail to the revised manuscript, stating “Participants also receive four doses of Infanrix-hexa, which is a popular choice of DTP-containing vaccine but is only available on the private market” (page 7).

Semi-major point 8: Page 12 of 33, Lines 16-19: “Other vaccinations are permitted in this study with a two- week interval from study vaccines.” Reviewer: This sentence is not clear. I read it as ‘Other EPI vaccinations are permitted but only if not given within two-weeks of receiving a study vaccine’

Response: We have clarified the wording in this sentence to read “With the exception of Quinvaxem, other vaccines are permitted in the study providing there are two weeks between the administration of other vaccines and study vaccines” (page 7)

Semi-major point 9: Page 12 of 33, Lines 39-48: Author writes: Traditional culture methods...will be the main methods used to analyse NP swabs collected at 6 and 12 months...qPCR targeting the *lytA* gene and serotyping by microarray will be the main methods used to analyse NP swabs collected at 18 and 24 months of age.” Reviewer: Please clarify why you are using more sensitive (microarray) and less specific (*lytA*) methodologies at the later visits. Clarify how you will address the fact that microarray increases likelihood of VT detection in co-carriage samples with NVT at high relative abundance and VT at low relative abundance (these samples would likely be defined as NVT using latex but NVT+VT in microarray).

Response: Due to funding constraints we were not able to perform DNA microarray for all time points. The 18 and 24 month samples are tested with a combination of *lytA* and microarray, with *lytA* effectively used as a screening test and the final identification achieved by the highly specific microarray. Although microarray increases the likelihood of VT detection in co-carriage samples, our traditional culture methods involve selection of any morphologically distinct colonies, reducing this difference. Furthermore, in our previous trial we found low rates of multiple serotype carriage by either method. Importantly, the same method is used for all groups at any given time point, so comparisons between groups will not be affected by any differences that remain.

Semi-major point 10: Page 12, Line 41: With reference to the use of *lytA* PCR, I would suggest including a comment on the limitation in using *lytA* PCR for pneumococcal detection (i.e. low specificity).

Response: The *lytA* PCR confirmation of nonencapsulated isolates was done in combination with traditional microbiological approaches for isolates (including optochin testing) and follows the WHO laboratory guidelines, as referenced in the appendix that includes more detailed laboratory methods.

Minor points

1. Page 4 of 33, Line 28-30; & page 9, line 22: refers to “...schedules as a strategy to maintain herd protection and to make PCVs more affordable. Reviewer: I think this refers to a less costly EPI programme (using 1-2 doses in place of 3-4) and not a less costly PCV vaccine. Please clarify.

Response: We have replaced “make PCVs more affordable” with “make pneumococcal vaccination

more affordable” (page 2 and page 4)

2. Page 5 of 33, Line 49/50: refers to “...PCV13 administered at 12 and 2 months...” Reviewer: Should likely read “...PCV13 administered at 2 and 12 months...”

Response: This has been corrected (the administrative information has been moved to a supplementary file)

3. Page 5 of 33, Line 55: reads “no significant maternal or perinatal history” Reviewer: The author should clarify to what type of history is being referred. I suspect it’s clinical history but please specify and/or add additional targeted conditions if relevant.

Response: This has been changed to specify clinical history (the administrative information has been moved to a supplementary file)

4. Page 11 of 33, Line 44-46: The author writes: “All vaccinations are performed by nurses specifically trained in infant vaccine administration. Reviewer: Please clarify if these are study-specific or MoH nurses.

Response: This has been clarified to read “All vaccinations are performed by Ministry of Health nurses specifically training in infant vaccine administration” (page 6)

5. Page 12 of 33, Lines 56-57: Author writes: “Serotype-specific pneumococcal carriage at 6, 12, 18 and 24 months of age” Reviewer: Is lab team able to determine all VT+NVT serotypes using latex (6 & 12m swabs)? Or should this be “VT serotype specific...”

Response: Yes, a full set of latex serotyping reagents is used, together with Quellung confirmation as required.

6. Page 18 of 33, Lines 16-26: Relevant to the section ‘Patient and public involvement’, is there any intention to disseminate results to the community, via townhall meetings, etc.?

Response: We are guided by our colleagues in Vietnam as to the most suitable communication methods and do not plan to disseminate results to the community via meetings etc at this stage

7. Page 20, Line 26: Relevant to the section ‘Authorship’, if feasible please clarify how members of the Publication Subcommittee will be selected.

Response: We have changed the wording from “publication subcommittee” to “small group of senior investigators”

VERSION 2 – REVIEW

REVIEWER	Swarthout, Todd UCL, Division of Infection and Immunity
REVIEW RETURNED	27-Oct-2021

GENERAL COMMENTS	Thank you for your timely response in addressing the reviewer's comments. I look forward to seeing the results. While acknowledging the question is not simple, I would argue that absence of PCV in a country's routine immunization programme does not in-of-itself justify investigating a reduced-dose schedule or
---

	use of a placebo arm. Though I don't feel it's absence should block the manuscript from being published, I would have liked to see a more nuanced discussion of that topic in a protocol manuscript.
--	--